# Involvement of Cellular Prion Protein in Invasion and Metastasis of Lung Cancer by Inducing Treg Cell Development

**DOI:** 10.3390/biom11020285

**Published:** 2021-02-15

**Authors:** Seunghwa Cha, Mi-Ji Sin, Mo-Jong Kim, Hee-Jun Kim, Yong-Sun Kim, Eun-Kyoung Choi, Mi-Yeon Kim

**Affiliations:** 1School of Systems Biomedical Science, Soongsil University, Seoul 06978, Korea; csh3@soongsil.ac.kr (S.C.); sinmiji1004@naver.com (M.-J.S.); 2Ilsong Institute of Life Science, Hallym University, Anyang 14066, Korea; hanbami0730@gmail.com (M.-J.K.); hijuni@himh.re.kr (H.-J.K.); yskim@hallym.ac.kr (Y.-S.K.); 3Department of Biomedical Gerontology, Graduate School of Hallym University, Chuncheon 24252, Korea

**Keywords:** prion, Treg cells, lung cancer, ME7, *Prnp^0/0^*, Tga20

## Abstract

The cellular prion protein (PrP^C^) is a cell surface glycoprotein expressed in many cell types that plays an important role in normal cellular processes. However, an increase in PrP^C^ expression has been associated with a variety of human cancers, where it may be involved in resistance to the proliferation and metastasis of cancer cells. PrP-deficient (*Prnp^0/0^*) and PrP-overexpressing (Tga20) mice were studied to evaluate the role of PrP^C^ in the invasion and metastasis of cancer. Tga20 mice, with increased PrP^C^, died more quickly from lung cancer than did the *Prnp^0/0^* mice, and this effect was associated with increased transforming growth factor-beta (TGF-β) and programmed death ligand-1 (PD-L1), which are important for the development and function of regulatory T (Treg) cells. The number of FoxP3^+^CD25^+^ Treg cells was increased in Tga20 mice compared to *Prnp^0/0^* mice, but there was no significant difference in either natural killer or cytotoxic T cell numbers. In addition, mice infected with the ME7 scrapie strain had decreased numbers of Treg cells and decreased expression of TGF-β and PD-L1. These results suggest that PrP^C^ plays an important role in invasion and metastasis of cancer cells by inducing Treg cells through upregulation of TGF-β and PD-L1 expression.

## 1. Introduction

The cellular prion protein (PrP^C^) is a glycosylphosphatidylinositol (GPI)-anchored cell surface glycoprotein widely expressed in various tissues but predominantly expressed in the central nervous system. PrP^C^ has also been detected on the surface of lymphocytes in humans and mice [1,2,3] and plays an important role in lymphoid cells and spleen structure formation [4,5,6]. Its wide expression in many cell types indicates diverse biological functions. While its exact function is unknown, PrP^C^ contributes to cell adhesion, neurite outgrowth, synaptic transmission, oxidative stress, cell death and survival, and signal transduction [7,8,9,10,11,12,13,14,15,16].

Cancer is a leading cause of death worldwide and metastasis is a major cause of death among cancer patients [17]. An increase in PrP^C^ expression has been associated with a poor prognosis and reported in a variety of human cancers, including pancreatic cancer, breast cancer, gastric carcinoma, osteosarcoma, and melanoma [18,19,20,21,22]. Studies to elucidate the mechanism have been performed by many groups through silencing or overexpressing PrP^C^ in human cancer cell lines. Gil et al. reported that PrP^C^ silencing in MD-MB231 breast cancer cell line led to ERK deactivation and matrix metalloprotease-9 (MMP-9) downregulation, resulting in decreased migration and invasion [19]. Besides, overexpression of PrP^C^ in MCF-7 cells activated the NF-κB and ERK pathway followed by increased MMP-9 expression [19]. Moreover, Wang et al. showed that PrP^C^ silencing or overexpression in pancreatic ductal adenocarcinoma modulated tumour invasion and proliferation by formation of the PrP^C^-Notch1 complex, resulting in increased Notch1 stability and activation [20]. In addition, PrP^C^-transfection into the AGS gastric cancer cell line upregulated anti-apoptotic protein Bcl-2 and downregulated p53 and Bax, resulting in slowed down apoptosis [22]. These results suggest that PrP^C^ is involved in increased survival, proliferation, and metastasis of human cancer cells.

The immune system is critical to controlling cancer cell proliferation and invasion. Regulatory T (Treg) cells are a subset of CD4 T cells and play a pivotal role in controlling immune responses by their immunosuppressive activity [23]. They are known to play an important role in the control of allergic inflammation and autoimmune responses [24,25], and increased numbers of Treg cells in cancer patients impede cancer immunotherapy [26,27,28]. Treg cells occur naturally in the thymus, called thymus-derived Treg cells, but are also induced in the periphery through exposure to antigens [24,25]. The key regulatory gene for the development of these Treg cells is a transcription factor, forkhead box P3 (FoxP3) [29], and the most important molecules for their suppressive activity are transforming growth factor-beta (TGF-β) and CD25, which is the IL-2 receptor subunit [30]. FoxP3-negative Treg cells were identified as type 1 Treg cells, which have immunosuppressive activity via high levels of IL-10 production [31]. Many studies in humans and mice have shown that anti-CD25, anti-CD4, or anti–CTLA-4 monoclonal antibody immunotherapies that inhibit the function of Treg cells or deplete them are successful [32,33,34,35,36].

We have previously reported that PrP^C^ is expressed in the secondary lymphoid tissues and has a critical role in the formation and maintenance of the white pulp structure in the spleen [4]. We also demonstrated that mouse-adapted scrapie strain ME7 infection led to an impaired white pulp structure, but follicular helper T cell responses were sustained to aid in the replication and accumulation of pathogenic PrP (PrP^Sc^) in germinal centres [6]. Thus, the suppressive action of the Treg cells is insufficient to control germinal centre responses when PrP^C^ has been converted into PrP^Sc^ in prion diseases. The role of PrP^C^ in the development of Treg cells in pathogenic conditions such as prion diseases and cancer is poorly understood. While the inhibition of the activity of Treg cells is a major target in immunotherapy treatments of cancer, the involvement of PrP^C^ in the development and induction of Treg cells has not been studied. Therefore, we investigated the role of PrP^C^ in the invasion and metastasis of cancer, via its induction of Treg cells, using Zürich I *Prnp*-deficient (*Prnp^0/0^*), wild-type C57BL/6J (*Prnp^+/+^*), and PrP overexpressing (Tga20) mice. Using the B16F10 melanoma cell system, we studied the metastasis of melanoma cells to the lungs and analysed the distribution of Treg cells and the expression of key genes related to their function in the spleen. We also compared these results to the data obtained from mice injected orally with the mouse-adapted scrapie strain ME7 to understand the effect of accumulation of PrP^Sc^ in the development and function of Treg cells.

## 2. Materials and Methods

### 2.1. Mice

Mouse strains Zürich I *Prnp^0/0^* [37] and Tga20 [38], which overexpressed mouse PrP, were kindly provided by Dr. A Aguzzi (Institute of Neuropathology, University Hospital of Zürich, Zürich, Switzerland) and Dr. C Weissmann (The Scripps Research Institute, Scripps Florida, Jupiter, FL, USA), respectively. The backgrounds of all mice are C57BL/6J (*Prnp^+/+^*), and the breeding stocks of C57BL/6J mice were originally purchased from the Jackson Laboratory (Bar Harbor, ME, USA). All mice were maintained as inbred strains in the animal facility of the Ilsong Institute of Life Science, Hallym University. The mice were housed in a clean facility with a 12 h light/12 h dark cycle and examined at 6–8 weeks of age.

### 2.2. Cancer Development

The *Prnp^0/0^*, *Prnp^+/+^*, and Tga20 mice were randomly divided into two groups: The negative control (NC) and the B16F10 melanoma cell injected (B16F10) groups (*n* = 4 for each group). B16F10 melanoma cells injected into the mouse tail vein resulted in a primary tumour around the injected area within 20 days. Circulating tumour cells in the blood enter the spleen, which is the secondary lymphoid organ and the main filter for blood-borne antigens [23,39]. If the antigens are not removed by the immune response and evade immune surveillance in the spleen, they invade through the endothelial cells of the blood vessel and reach other organs, commonly the lungs in the case of melanoma cells. Approximately three weeks after injection, the B16F10 cells implant, grow, and form metastatic foci in the lungs [17]. The NC group was injected with 200 μL phosphate-buffered saline (PBS), and the B16F10 group was injected intravenously with 5 × 10^5^ B16F10 melanoma cells in 200 μL PBS to induce the development of lung cancer. These experiments, using the same number of mice, were repeated three times (total *n* = 12 for each group). B16F10 cells were purchased from the Korean Cell Line Bank (Seoul, Korea) and cultured in DMEM containing 10% FBS, 0.1 mg/mL streptomycin, and 100 U/mL penicillin at 37 °C under 5% CO_2_. After injection, the mice were observed several times daily for clinical signs of cancer. When the mice showed the terminal stage symptoms, which included weight loss, unresponsiveness to stimuli, and lack of movement, they died within 24 h. After the mice were sacrificed, the lungs were examined for evidence of metastasis (Figure 1A), and the spleen was collected for further analysis.

### 2.3. Scrapie Infection

The original stock of the ME7 scrapie strain was kindly provided by Dr. Alan Dickinson of the Agriculture and Food Research Council and Medical Research Council Institute (Neuropathogenesis Unit, Edinburgh, UK). For scrapie infection, 8 wild-type C57BL/6J mice were divided into two groups: The control and ME7-infected groups (*n* = 4 for each group). The ME7-infected group was orally infected by gavage with 100 μL of 1% brain homogenate from wild-type mice infected with the mouse-adapted ME7 scrapie strain. The mice were observed several times each week for clinical signs of scrapie, which included weight loss, kyphosis, and ataxia. The terminal stage of scrapie was reached in about 260 days. These experiments, using the same number of mice, were repeated three times (total *n* = 12 for each group).

### 2.4. Quantitative PCR Analysis

mRNA was isolated using the Ribospin system (Geneall, Seoul, Korea) according to the manufacturer’s recommendations. cDNA preparation and quantitative real-time PCR (Takara, Shiga, Japan) were performed as previously described [40,41]. The relative expression of the *Actb* control was calculated as 2^−ΔCt^ × 10^2^. Expression of each target was normalised to *Actb* expression (*Actb* = 100%). The primer sequences were synthesised by Bioneer (Daejeon, Korea) and are listed in Table 1.

### 2.5. Confocal Images

Immunofluorescence and confocal microscopy imaging were performed as described previously [4]. The primary antibodies used in our analyses were: FITC-conjugated anti-mouse FoxP3 (FJK-16s, eBioscience, San Diego, CA, USA), APC- conjugated anti-mouse CD3e (145-2C11, eBioscience), and rhodamine-conjugated anti-mouse IgM (Jackson ImmunoResearch, West Grove, PA, USA). Sections were mounted using ProLong Gold Antifade Reagent (Molecular Probes, Carlsbad, CA, USA), and confocal images were obtained using an LSM 700 Meta microscope (Carl Zeiss, Oberkochen, Germany), equipped with 488, 568, and 633 nm lasers. Images were analysed using Zeiss LSM microscopy software Zen 2.6 (Carl Zeiss).

### 2.6. Flow Cytometry

Monoclonal antibodies for CD25 (PC61.5) and FoxP3 (150D/E4) were purchased from eBioscience (San Diego, CA, USA). Anti-CD3e (145-2C11), anti-CD4 (RM4-5), anti-CD279 (PD1) (J43), anti-CD8a (53-6.7), and anti-CD49b (DX5) antibodies were purchased from BD Biosciences (San Jose, CA, USA). Data were collected with FACSCaliber (BD Biosciences) and analysed using Flowjo software 8.7 (TreeStar, San Carlos, CA, USA).

### 2.7. Statistical Analysis

For all experiments, significance was calculated using the Student’s *t*-test and one-way analysis of variance (ANOVA), with *p* < 0.05 considered significant; * *p* < 0.05 and ** *p* < 0.01, as compared to the control.

## 3. Results

### 3.1. Effect of PrP^C^ Expression on Lung Cancer Development

To evaluate the role of PrP^C^ in invasion and metastasis of lung cancer, we intravenously injected B16F10 melanoma cells into *Prnp^0/0^*, *Prnp^+/+^*, and *Prnp*−overexpressing (Tga20) mice. When the mice reached the terminal stage of cancer, the lungs were taken and examined for melanoma cell invasion, indicating that metastasis had occurred (Figure 1A). Compared to *Prnp^0/0^* mice, which required 31.4 days to reach the end-stage disease, *Prnp^+/+^* and Tga20 mice required 24.6 and 23.7 days, respectively (Figure 1B). Basal level of *Prnp* expression compared to *Prnp^0/0^* mice was 7.2% in *Prnp^+/+^* and 88.0% in Tga20 mice (Figure 1C). In addition, the *Prnp* levels at the end-stage of cancer were upregulated in both *Prnp^+/+^* and Tga20 mice (Figure 1C). These results indicate that PrP^C^ expression is associated with survival time and lung cancer development.

### 3.2. Increased Numbers of Treg Cells in Prnp^+/+^ and Tga20 Mice

Since Treg cells are elevated in cancers and suppress anti-tumour immune response [26,27,28,42], we analysed the distribution of Treg cells in spleens from mice that had B16F10 melanoma cells (Figure 2). At the terminal stage of cancer, FoxP3-expressing Treg cells were found in white pulp from all three strains (Figure 2B,D,F), but Tga20 mice showed increased numbers of the cells (Figure 2F(f-1)). As previously reported, *Prnp^0/0^* mice showed impaired T and B cell zone structure [4] but did not show a significant difference in Treg cell numbers before and after B16F10 cell injection (Figure 2A,B). The number of Treg cells was determined by flow cytometric analysis of FoxP3 and CD25 expression (Figure 3A). In the *Prnp^0/0^* mice, there was no difference in the ratio of FoxP3 and CD25 expressing CD4 T cells to all CD4 T cells between the NC mice and the B16F10 mice in the terminal stage of cancer. In the *Prnp^+/+^* mice, the ratio of FoxP3 and CD25 expressing CD4 T cells was 15.0% in the NC mice and 21.0% in the B16F10 mice, while the ratio for Tga20 mice was 18.5% in the NC mice and 27.3% in the B16F10 mice (Figure 3A,B).

### 3.3. Increased Expression of TGF-β and PD-L1 in Tga20 Mice

Expression of *Tgfb*, which is required for Treg cell activity, increased in *Prnp^+/+^* and Tga20 mice from 9.4% to 11.9%, and 9.5% to 12.2%, respectively, in the B16F10 mice compared to the NC mice (Figure 3C), which indicates that PrP^C^ may have an important role in the development of Treg cells. While the Treg cell numbers in *Prnp^0/0^*, *Prnp^+/+^*, and Tga20 mice in the NC group did not show significant differences, PrP^C^ may have a role in inducing Treg cells after antigen stimulation. Analysis of the expression of programmed death ligand-1 (*Pdl1*), a regulator of the development, maintenance, and function of induced Treg cells [43], revealed similar levels in all NC mice; however, *Pdl1* was upregulated from 16.8% to 21.7% in Tga20 mice in the B16F10 mice compared to the NC mice (Figure 3D). The expression of programmed cell death protein 1 (PD1), a receptor for PD-L1, was upregulated 1.5- and 1.6-fold, respectively, in both *Prnp^+/+^* and Tga20 B16F10 mice relative to the NC mice (Figure 3E). The expression of another important cytokine, IL-10, did not show any significant differences among groups (data not shown). These results suggest the possible involvement of PrP^C^ for the induction of peripherally derived FoxP3^+^CD25^+^ Treg cells through upregulation of TGF-β and PD-L1 expression.

### 3.4. No Effect of PrP^C^ Expression on Natural Killer or CD8 T Cell Numbers

Natural killer (NK) and CD8 T cells, which play a pivotal role in anti-cancer immunity [44], were analysed by flow cytometry but did not show differences among *Prnp^0/0^*, *Prnp^+/+^*, and Tga20 mice (Figure 4A,B). NK cells were slightly decreased in all of the B16F10 mice compared to NC mice (Figure 4A), but the small changes between the mice groups for CD8 T cells were not statistically significant (Figure 4B).

### 3.5. Decreased Numbers of Treg Cells by introducing PrP^Sc^ through ME7 Scrapie Infection

We previously reported that mice inoculated intraperitoneally with the mouse- adapted scrapie strain ME7 have significantly diminished T cell zones and increased follicular helper T cell responses in the spleen [6]. Because PrP^Sc^ is frequently obtained from food, we infected mice with ME7 orally and investigated the spleen structure and Treg cells (Figure 5). At the terminal stage of the prion disease caused by ME7 infection, the spleen had an impaired white pulp structure (Figure 5A(c,d)), similar to the spleens of mice inoculated intraperitoneally with ME7 [6]. The spleens of ME7-infected mice showed decreased numbers of Foxp3-expressing Treg cells: 8.5% in the control and 4.2% in the ME7-infected mice (Figure 5A,B). The expression levels of Tgfb, Il10, and Pdl1 also decreased after ME7 infection: For Tgfb, from 9.9% in the control to 7.1% in ME7-infected mice; for Il10, from 3.2% to 2.2% after infection; and for Pdl1 from 2.7% to 1.7% after infection (Figure 5C). From these results, ME7-infected mice had decreased numbers of Treg cells and decreased expression of key genes related to their function.

## 4. Discussion

We investigated the role of PrP^C^ in the development and function of Treg cells in cancer using the B16F10 melanoma cell system to induce lung metastasis in *Prnp^0/0^*, *Prnp^+/+^*, and Tga20 mice. *Prnp^+/+^* and Tga20 mice reached end-stage cancer more quickly (about 24 days) than did *Prnp^0/0^* mice (about 31 days) suggesting that PrP^C^ promoted invasion and metastasis of lung cancer; however, the amount of PrP^C^ did not affect the time it took for mice to reach end-stage cancer. These results support the hypothesis that PrP^C^ promotes the invasion and metastasis of lung cancer; however, the survival time of the mice did not correlate with the amount of PrP^C^.

Since the relationship between PrP^C^ and cancer progression was first reported in pancreatic cancer cells [45], increasing evidence has been suggesting the involvement of PrP^C^ in tumourigenesis through interacting and activating binding partners and signaling pathways [46]. Moreover, PrP^C^ in the extracellular space, due to proteolytic cleavage or lack of glycosylation, has been found to complex with anthracycline followed by reducing its chemotherapeutic activity to a large extent [47]. PrP^C^ has been discovered to be present in a Pro-PrP form, neither having the GPI anchor nor being glycosylated in different human cancer cell lines and tissues [46,48,49]. Interestingly, the Pro-PrP isoform continues to possess the GPI anchor peptide signal sequence which is able to provide a binding motif for filamin A, which participates in modulating cancer cell proliferation and migration in colorectal cancer cells and the expression of BRCA1 in breast cancer [48,50,51]. However, the involvement of PrP^C^ itself or its various forms in different vital pathways that control cancer cell migration, tumour invasion and metastasis, and immune responses during cancer development still needs to be elucidated.

Treg cells have an important role in immune response through their immunosuppressive activity, and high numbers of Treg cells impede cancer immunotherapy [26,27,28]. We determined the effect of PrP^C^ on Treg cells and the time to metastasis. At the end-stage of cancer, Treg cell numbers had increased about 1.5-fold in B16F10-treated *Prnp^+/+^* and Tga20 mice compared to untreated control mice; however, there was no increase in *Prnp^0/0^* mice. PrP^C^ seems to be essential for the induction of peripherally derived Treg cells; however, since the untreated *Prnp^0/0^*, *Prnp^+/+^*, and Tga20 control mice all had similar numbers of Treg cells, PrP^C^ is not required for thymus-derived Treg cell development. We measured the expression of TGF-β and PD-L1, which are critical for inducing peripherally derived Treg cells, and found that TGF-β expression was increased in *Prnp^+/+^* and Tga20 mice after B16F10 treatment, while PD-L1 expression was upregulated only in Tga20 mice. This suggests that PrP^C^ promotes peripherally derived Treg cell development after antigen exposure through increased expression of TGF-β (independent of the amount of PrP^C^) and induces more Treg cells through PD-L1 upregulation (which is dependent on the amount of PrP^C^). However, PrP^C^ is not involved in the development of thymus-derived Treg cells.

Expression of PD1, a receptor for PD-L1, was highly upregulated on CD3 T cells in *Prnp^+/+^* and Tga20 mice, but not in *Prnp^0/0^* mice. PD1 is a negative regulator of T cell activity [52,53], and its interaction with PD-L1 on normal cells prevents overstimulation, mediated by activated T cells; however, its interaction with PD-L1 on cancer cells suppresses the anti-tumour effector activity of the T cell [54]. Therefore, the upregulation of PD1 on T cells in *Prnp^+/+^* and Tga20 mice and its interaction with PD-L1 on B16F10 melanoma cells, which express PD-L1 [55], could help melanoma cells to evade immune surveillance, thereby facilitating invasion and metastasis.

Our previous study showed that infection with the mouse-adapted scrapie strain ME7 led to an impaired splenic white pulp structure, but follicular helper T cell responses were prolonged to aid in the replication and accumulation of PrP^Sc^ [6]. When cellular PrP^C^ has been converted to create an abundance of PrP^Sc^, the suppressive action of the Treg cells cannot control germinal centre responses. In this study, we also analysed the spleens of *Prnp^+/+^* mice infected orally with ME7 to determine whether Treg cells were decreased at the end stage of prion disease when PrP^C^ had been converted to PrP^Sc^. In agreement with previous studies in which mice were infected with ME7 intracerebrally [5] or intraperitoneally [6], mice infected orally with ME7 showed impaired white pulp structure and disturbed segregation of T and B cell zones. Treg cells were decreased in the spleens of ME7 infected mice, as were the expression levels of TGF-β, IL-10, and PD-L1. Therefore, the reduced Treg cells were insufficient to suppress follicular helper T cell responses, which promoted the replication and accumulation of PrP^Sc^ in ME7-infected mice. These results, along with those from the cancer metastasis study, support the important role of Treg cells in lung cancer invasion and metastasis processes, whose growth is associated with expression of PrP^C^, which enhances cancer progression.

## 5. Conclusions

This study provides insights into the role of PrP^C^ in the development and function of Treg cells. During the invasion and metastasis stages of cancer progression, increased expression of PrP^C^ induces the development of Treg cells via the upregulation of TGF-β and PD-L1, thereby accelerating cancer progression by suppressing immune response. In contrast, the pathogenic state of prion diseases inhibits the development of Treg cells via the downregulation of TGF-β, IL-10, and PD-L1, clearly correlating the involvement of PrP^C^ in Treg cell development. Taken together, our results have revealed a novel role of PrP^C^ in the development and function of Treg cells that may be a future target for additional therapeutics in cancer treatment.

## Figures and Tables

**Figure 1 biomolecules-11-00285-f001:**
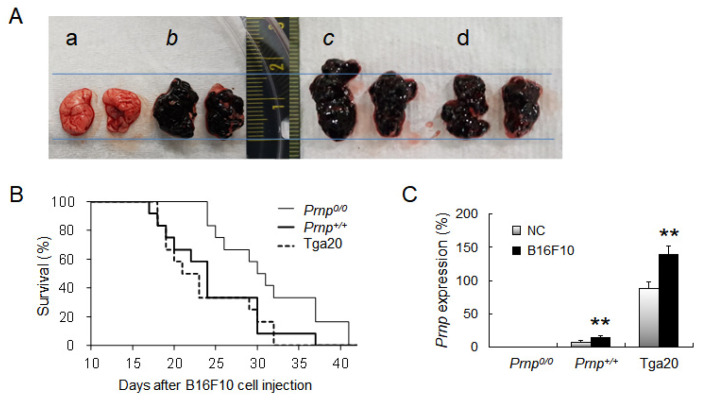
The terminal stage of lung cancer induced by B16F10 melanoma cells of *Prnp^0/0^*, *Prnp^+/+^*, and Tga20 mice. (**A**) Representative images of tumours. (a) Negative control. (b–d) The lungs at the terminal stage of cancer from *Prnp^0/0^* (b), *Prnp^+/+^* (c), and Tga20 (d) mice. (**B**) Survival curve for mice with metastatic lung cancer. (**C**) mRNA levels for *Prnp* in whole splenocytes. NC: Negative control group; B16F10: Group with lung cancer induced by B16F10 melanoma cells. mRNA expression was normalised to *Actb* expression (100%). These results are for an average of 12 mice per group. Error bars represent mean values ± standard deviation. ** *p* < 0.01.

**Figure 2 biomolecules-11-00285-f002:**
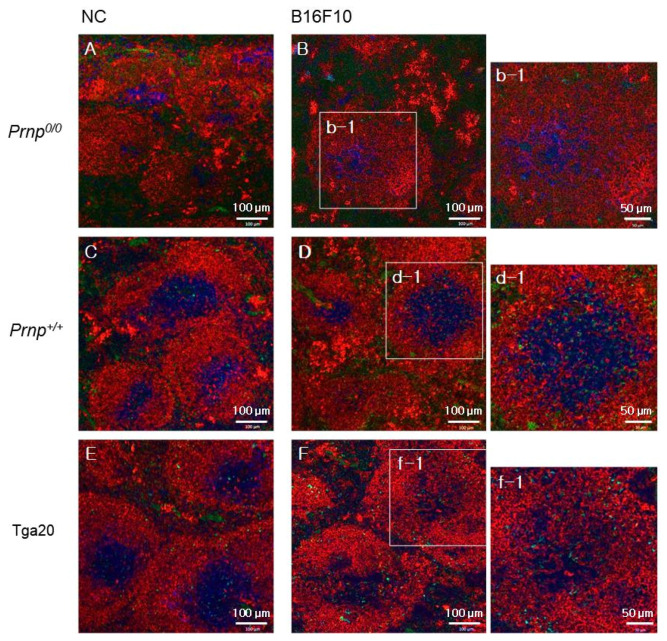
Low power confocal micrographs of *Prnp^0/0^* (**A**,**B**), *Prnp^+/+^* (**C**,**D**), and Tga20 (**E**,**F**) mouse spleen sections showing white pulp. Images show CD3 (blue for T cells), IgM (red for B cells), and FoxP3 (green for Treg cells). (**A**,**C**,**E**) The spleens from normal mice (negative control, NC) and (**B**,**D**,**F**) from B16F10-treated mice (B16F10) at the terminal stage of lung cancer. High-power confocal images (b-1, d-1, and f-1) are from the area in the white boxes in B, D, and F. Data are representative of sections from three mice per group.

**Figure 3 biomolecules-11-00285-f003:**
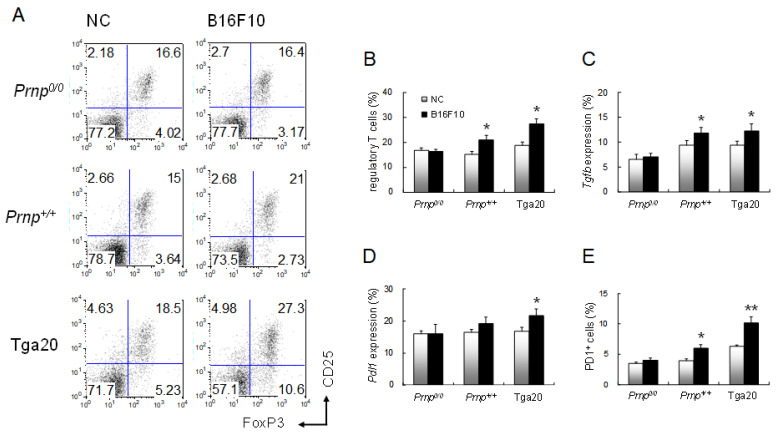
Analyses of Treg cells in the spleens of the negative control (NC) and B16F10-treated groups at the terminal stage of lung cancer. (**A**) Flow cytometric analysis of FoxP3 and CD25 expression in CD4^+^CD3^+^ gated cells. (**B**) The number of Treg cells calculated from flow cytometric plots. (**C**) mRNA levels for *Tgfb*. (**D**) mRNA levels for *Pdl1*. Relative mRNA levels were measured by quantitative real-time PCR and normalised to *Actb* expression (100%). (**E**) Flow cytometric analysis of CD3^+^PD1^+^ cells (%). Data are for an average of 12 mice per group. Error bars represent mean values ± standard deviation. * *p* < 0.05 and ** *p* < 0.01.

**Figure 4 biomolecules-11-00285-f004:**
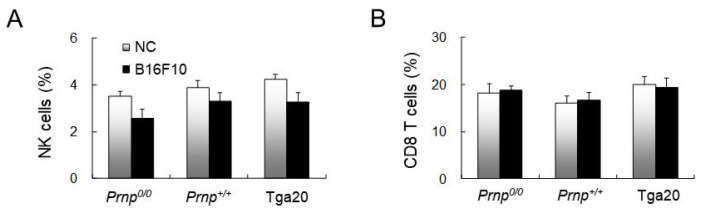
Flow cytometric analysis of cancer-related cells in the spleens of the negative control (NC) and B16F10-treated groups at the terminal stage of lung cancer. (**A**) DX5^+^ NK cells (%). (**B**) CD3^+^CD8^+^ cells (%). Data are for an average of 12 mice per group. Error bars represent mean values ± standard deviation.

**Figure 5 biomolecules-11-00285-f005:**
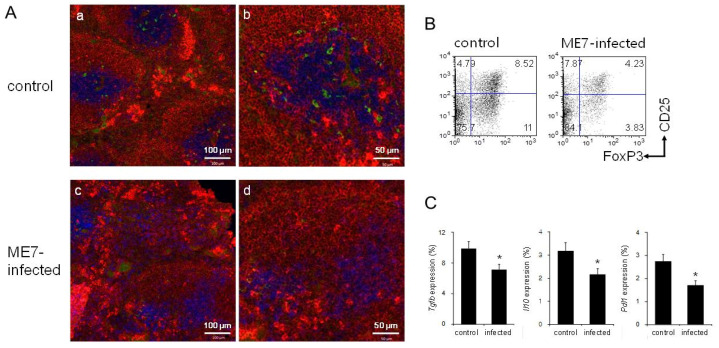
Analyses of FoxP3^+^ Treg cells from control and ME7-infected mouse spleens. (**A**) Confocal images of spleen sections: (**a**,**c**) Low-power images of white pulp, (**b**,**d**) high-power images. Immunofluorescent staining indicates CD3 (blue for T cells), IgM (red for B cells), and FoxP3 (green for Treg cells). (**a**,**b**) Control spleen. (**c**,**d**) ME7-infected spleen. Data are representative of sections from three mice per group. (**B**) Flow cytometric analyses of CD25 and FoxP3 expression by CD3^+^CD4^+^ gated cells. (**C**) mRNA levels for *Tgfb*, *Il10*, and *Pdl1*. Relative mRNA levels were measured by quantitative real-time PCR and normalised to *Actb* expression (100%). Data represent the average expression of 12 mice per group. Error bars represent mean values ± standard deviation. * *p* < 0.05.

**Table 1 biomolecules-11-00285-t001:** List of primer sequences for quantitative PCR analysis.

Gene	Forward Primer	Reverse Primer
*Actb*	CGTGAAAAGATGACCCAGATCA	TGGTACGACCAGAGGCATACAG
*Prnp*	ATGGCGAACCTTGGCTACTG	CCTGAGGTGGGTAACGGTTG
*Tgfb*	CCGCAACAACGCCATCTATG	CCCGAATGTCTGACGTATTGAAG
*Pdl1*	GCTCCA AAGGACTTGTACGTG	TGATCTGAAGGGCAGCATTTC
*Il10*	ACAGCCGGGAAGACAATAACT	GCAGCTCTAGGAGCATGTGG

## Data Availability

Data sharing not applicable.

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
