# Peer review of "Involvement of Cellular Prion Protein in Invasion and Metastasis of Lung Cancer by Inducing Treg Cell Development"

_biomolecules, 2021, doi:10.3390/biom11020285_

Round 1
Reviewer 1 Report
The manuscript covers an interesting topic describing the influence and role of prion protein on lung cancer invasion and metastasis. The authors demonstrate the important involvement in these processes of Treg Cells, whose growth is associated with high expression of PrP, which enhances cancer progression.
I provide minor comments below:
The introduction and the discussion should have been slightly more extended.
The authors should emphasise novelty even more clearly.
The authors should add a list of abbreviations.
I am curious about the authors' opinion as to why there was no effect of PrP expression on Natural Killer or CD8 T Cell numbers?
Reviewer 2 Report
In this paper Seunghwa Cha and colleagues investigated the role of cellular prion protein (PrPC) in promoting cancer cell invasion and metastasis through the regulation of Treg development.
Overall, the manuscript is well written and the detailed data shed new lights on the possible interpretation regarding the involvement of PrPC in the pathophysiology of cancer progression and metastasis.
Within this frame, it would be interesting to know if the Authors have considered to extend the discussion regarding other PrP isoforms and/or PrP fragments. This is the case, for instance, of pro-PrP isoform, which has been recently discovered in human pancreatic ductal adenocarcinoma (PDAC) and melanoma cell lines. Being neither glycosylated nor GPI-anchored, pro-PrP retains the GPI anchor peptide signal sequence (GPI-PSS) which contains a filamin A binding motif, thus providing a growth advantage in these cell lines (Binding of pro-prion to filamin A disrupts cytoskeleton and correlates with poor prognosis in pancreatic cancer. Li C, Yu S, Nakamura F, Yin S, Xu J, Petrolla AA, Singh N, Tartakoff A, Abbott DW, Xin W, Sy MS. J Clin Invest. 2009 Sep; 119(9):2725-36; Pro-prion binds filamin A, facilitating its interaction with integrin beta1, and contributes to melanomagenesis. Li C, Yu S, Nakamura F, Pentikäinen OT, Singh N, Yin S, Xin W, Sy MS. J Biol Chem. 2010 Sep 24; 285(39):30328-39; The Role of Cellular Prion Protein in Promoting Stemness and Differentiation in Cancer. Ryskalin L, Biagioni F, Busceti CL, Giambelluca MA, Morelli L, Frati A, Fornai F. Cancers (Basel). 2021 Jan 6;13(2):170.).
A minor issue is to indicate “PrPC” when referring to “cellular prion protein” through the manuscript.
